# Effect of Age and Gender on the Efficacy of a 12-Month Body Weight Reduction Program Conducted Online—A Prospective Cohort Study

**DOI:** 10.3390/ijerph191912009

**Published:** 2022-09-22

**Authors:** Jakub Woźniak, Katarzyna Woźniak, Olga Wojciechowska, Michał Wrzosek, Dariusz Włodarek

**Affiliations:** 1Department of Dietetics, Institute of Human Nutrition, Warsaw University of Life Sciences (WULS–SGGW), Nowoursynowska 159 C, 02-776 Warsaw, Poland; 2Centrum Respo, Chmielna 73, 00-801 Warsaw, Poland

**Keywords:** obesity, overweight, human, gender, men, women, energy restriction, physical activity, lose weight, online intervention

## Abstract

Overweight and obesity are a cause of many non-communicable diseases leading to an increased risk of death. There are many programs aimed at weight reduction, but few publications have evaluated their effectiveness according to the gender and age of the subjects. The purpose of this study is to evaluate the effects of age and gender on weight loss outcomes in subjects participating in a 12-month online weight loss program. 400 subjects, 190 men and 210 women, were included in the study. The online intervention consisted of a 15% energy deficit diet and training (RESPO method). Changes in body weight over 12 months were similar (*p* = 0.14) across age groups. Weight reductions by month were statistically significant (*p* = 0.0001) in both groups. We noted no differences in weight loss between men and women expressed in kilograms. However, women reduced their body weight to a greater extent, i.e., by 2.7 percentage points, than men. Gender is a factor that may influence the effectiveness of weight loss programs, while age demonstrates no such influence. Our study shows that significant weight reduction during weight loss therapy is achieved by both men and women, but women can expect better results.

## 1. Introduction

The number of adults with an abnormal body mass index (BMI) is escalating globally, with more and more people being categorised as overweight or obese. Moreover, the prediction models reckon that this shift in BMI will continue. Approximately, in the European Union (EU) 36% of people are considered overweight and 17% obese, which adds up to around 53% being considered above the recommended weight [1]. Obesity is a global concern as a risk factor for numerous diseases. Excessive fat accumulation in body is positively correlated with increased risk of diabetes type 2, cardiovascular diseases, musculoskeletal and kidney disorders and several types of cancer, including breast, ovarian, prostate, liver, gallbladder and colon [2]. In the United Kingdom only, it was estimated, that over GBP 6.1 billion were spent on obesity-related treatments from 2014 to 2015 [3]. What is more, the recent COVID-19 pandemic seems to escalate the prevalence of obesity. Prolonged lockdowns, extended stress and closure of indoor sport facilities could result in increased energy intake [4]. Moreover, obese individuals appear to have higher rates of hospitalization, intensive care unit (ICU) admissions and mortality after being infected with SARS-CoV-2 [5]. Ignored and untreated, obesity can lead to serious health issues or could be a cause of disability or even death. Therefore, it is crucial to understand and detect obesity to prevent and decrease its prevalence.

According to statistics, in the EU there is no systematic difference between the genders as regards the share of obese women and men [1]. However, men seem to be underrepresented in weight loss programs. Numerous weight-loss interventions target mostly women. Furthermore, social norms and negative body image could be the cause why women engage with dietary programs more frequently [6]. Despite that, it has been argued that males are more successful with weight loss during dietary interventions [7]. That could be caused by different fat distribution among genders and lower average fat mass percentage in men, compared to women [8]. Age has been positively correlated with excessive fat mass. In the EU, the percentage of overweight individuals is increased with age [1]. The age group “18–24” has the lowest prevalence of overweight individuals (25%), while in the age group “65–74” almost 66% had been classified as overweight. Studies suggest a broad variation in weight gain over time in different populations. In western countries weight increase among adults was estimated between 200 and 400 g/year [9]. Therefore, it is important to target the prevalence of obesity. It is estimated that even moderate weight loss of 5–10% of the baseline weight is sufficient for health improvements [10].

## 2. Methodology

### 2.1. Study Design and Purpose

This study was designed as an observational, prospective, open-labeled, twelve-month trial. Data collected concerned the weight loss program from January 2019 to December 2021. This report is an extension of the observations made and described in the previously published article [11]. In the previous work, we focused on the analysis of the results in terms of the impact of BMI on the rate of weight loss. To find out more about the methodology of the conducted research, we encourage you to read the quoted article [11]. Below we have focused on the most important aspects of the methodology. The purpose of this study is to evaluate the effects of age and gender on weight loss outcomes in subjects participating in a 12-month online weight loss program. We focused on assessing weight changes over the course of the program in relative terms (percent reduction in baseline body weight) rather than in absolute terms (in kilograms). Describing weight loss in this way allows to look at the phenomenon in a more objective way.

### 2.2. Sample

720 overweight and obese subjects were recruited for the study. 400 subjects completed the protocol and was included in this analysis. The mean BMI in the group was 31.83 ± 4.77 (min 25.1 max 51.8). Men constituted 190 subjects and women 210 subjects. Inclusion criteria were overweight subjects between 18 and 55 years of age, no dietary intervention in the last 24 months, participation in a dietary intervention for 12 months, no musculoskeletal injuries, accessibility to a computer and/or telephone, no physician contraindication to regular physical activity.

### 2.3. Study Procedure and Intervention Characteristics

Data sources included nutrition and medical forms completed by subjects prior to the nutrition intervention, anthropometric measurements at baseline and taken during the intervention [12].

The online intervention consisted of a 15% energy deficit diet and training (RESPO method). Respo method (from the word “responsiveness”) is based on individual adjustment of the whole plan of losing weight. The Respo method is based on 4 fundamentals. The first is an individually tailored diet in terms of, among other things, taste preferences, number of meals or meal times. The second aspect of the method is a training plan adapted to the participants. The third foundation is constant contact with an online dietician, so that the participant could contact him at any time. The final aspect of this method is the emphasis on acquiring healthy eating habits. In the process of dietitian-participant collaboration, efforts were made to educate the participants as often as possible.

The intervention diet was designed according to the recommendations for healthy adults in Poland [13]. The energy value of the diet was reduced by 15% relative to the participants’ TDEE (total daily energy expenditure), which was determined based on BMR (basal metabolic rate) estimated using the Harris and Benedict formula, including the physical activity index (PAL) as recommended by the Institute of Food and Nutrition in Poland. [13]. The level of physical activity was assessed by using the physical activity questionnaire published by Johansson and Westerterp [14]. 

The PAL coefficient oscillated between 1.2 and 2. The proportion of carbs in the diet was set at 50–55% of the energy value of the diet, with simple sugars added less than 10%. The proportion of energy from fat was 25–35% of the energy value of the diet. Protein, meanwhile, was set at 1.6 g of protein per kilogram of participants’ body weight. [15]. The diet was balanced based on a program having the Food and Nutrition Institute and USDA product and food database. The general diet characteristics are presented in Table 1.

### 2.4. Outcome Measurements

The measurements of height and weight were taken by the subjects themselves after a pre-prepared training session in which a specialized dietician demonstrated how to perform these steps correctly. Each subsequent measurement was taken by the subjects at home on an empty stomach every 14 days to minimize measurement error. Measurements were taken from home to make the entire intervention process as easy as possible for the participants. During the dietary intervention, participants were in constant contact with the dietitian and trainer and submitted bi-weekly reports including body weight. All participants in the study had contact with a dietitian at least every 2 weeks to ensure the same model of intervention for each person. In addition, they completed a running diary each day to monitor the actual intake of the intended diet. Each study participant received appropriate instructions for completing the food diary.

### 2.5. Statistical Analysis

STATISTICA 13.3 PL package (TIBCO Software Inc. (2017). Statistica (data analysis software system), version 13 was used to elaborate the data obtained. For all statistical analysis, a level of *p* < 0.05 was taken as the cut-off for rejection of the null hypothesis. Basic descriptive statistics were calculated for quantitative data and distributions of qualitative characteristics were determined using multivariate (contingency) tables. The significance of differences in the distribution of qualitative characteristics was tested using the chi^2 test in combination with the multivariate tables. Due to the rejection for most of the analyzed variables by the W Shapiro–Wilk test of the hypothesis of normality of distribution and the expression of a significant part of the variables in ordinal scales, non-parametric tests were used in the study—the Mann–Whitney U test (with correction for continuity), the Wilcoxon signed-rank test and ANOVA—Kruskal–Wallis or Friedman with post hoc tests, respectively.

## 3. Results

Immediately prior to the dietary intervention, the mean BMI of subjects participating in the weight reduction program was 31.83 ± 4.77. The mean age was 33.42 ± 7.2 years. Physical activity as measured by PAL was at a moderate level and was 1.49 ± 0.15. The general characteristics of the subjects at the beginning of the intervention are shown in Table 2. 

The numbers of subjects after splitting into groups were as follows: 190 men and 210 women. The study groups did not differ statistically significantly in age, BMI values, physical activity index (PAL), total time spent training per week. Men were taller, had higher absolute body weight, higher basal and total metabolism than women. Additionally, women trained more frequently (*p* = 0.01) than men on a weekly basis; on the other hand, their time per training unit was significantly shorter (*p* = 0.02), it was 55 ± 17.99 min on average compared to 58.7 ± 17.96 min for men. The characteristics of the men and women participating in the study at the beginning of the intervention are presented in Table 3. 

The number of 18–29-year olds was 136, 30–39-year olds was 156 and 40–55-year olds was 108. There were no significant differences in the proportion of men and women in all 3 groups (*p* = 0.33). The groups did not differ in BMI, basal metabolism, total metabolism and training time both in terms of time per training unit and time spent training per week. Factors differentiating the groups, in addition to age, were absolute body weight, physical activity index, and number of trainings per week. Characteristics of all subjects observed at the beginning of the intervention divided into 3 age groups are presented in Table 4.

In both groups after 12 months of intervention, the absolute reduction in body weight from month to month was significant (*p* = 0.0001) and was for men between 0.7 kg and 2.2 kg which was reflected in a relative weight reduction of between 0.9% and 2.2% each month. Women, on the other hand, reduced their body weight between 0.7–1.8 kg per month which in relative terms means a reduction of 0.9–2% in each subsequent month. During the 12 months of the program, men reduced their body weight by a total of 15.2 kg which means a relative weight reduction of 14.7 %. In contrast, women lost weight by 15.4 kg and due to their lower initial body weight, this was reflected in a relative weight reduction of 17.4%. Women reduced their body weight at 12 months to a greater extent, i.e., by 2.7 percentage points at *p* = 0.00001 than men (Mann–Whitney U Test). Interestingly, in the first 3 months, men lost weight at a similar rate as women, but from the 4th to the 11th month the weight loss in their group was already lower than in the women’s group. General changes in body weight in gender groups after 12 months are shown in Figure 1a,b. Changes in body weight in gender groups in every month are shown in Appendix A. 

Changes in body weight across the 3 age groups over 12 months are shown in Figure 2. The relative weight reduction over 12 months was similar (*p* = 0.14) across age groups at approximately 16% (Kruskall-Wallis test). Person in age 18–29 reduced 16.5% of body mass, person in age 30–39 reduced 16% of body mass, person in age 40–55 reduced 15.9% of body mass. In other words, age did not affect the effectiveness of weight reduction in those participating in the online program. Similar to the gender groups, weight loss occurred in each month of the intervention and was significant relative to the previous month (*p* = 0.0001). More information about the changes in body weight across the 3 age groups in every month are presented in Appendix A.

## 4. Discussion

In this study, we addressed the determination of the effect of gender on the magnitude of weight reduction during a therapeutic process conducted remotely via online communication. A very strong element of our study is the participation of a similar number of women and men (men constituted 47.5% of the group). Additionally, in the analysis of the different age groups, gender was not a differentiating determinant and so in all 3 groups the proportion of women and men was similar. This allowed to assess how age affects the rate of excess weight reduction without the important differentiating factor of gender. In a systematic review by Pagoto et al. [16], the researchers found that in the studies included in the review men accounted for only 27% of the total study population, additionally it is worth noting that this percentage was only slightly higher for obesity interventions with associated comorbidities (36% men). In another study targeting weight reduction, men made up only 20% of the study participants [17]. There are many hypotheses explaining the fact that men are less representative in studies evaluating the effectiveness of methods to reduce excess body weight. It is possible that women have a greater desire to change their body weight due to cultural pressures regarding their appearance [18]. On the other hand, men are culturally encouraged to maintain more muscle mass and therefore more body weight [19]. Still another hypothesis that is raised in the realm of relating the topic of weight loss is that socially the use of dieting is stereotypically associated with women [20,21]. Therefore, this study is unique in the literature on the topic in that a similar sized group of women and men were assessed. It also counters the general trend that men are less likely to participate in weight loss programs. 

Interestingly, the available literature reviews are contentious in clearly defining whether gender affects the results obtained during weight loss. A systematic literature review from 2011 reported that such differences were not observed in the analyzed studies [22]. Similar observations are noted in studies on the effect of regular physical activity on the rate of weight loss. In a 2013 study lasting 10 months, subjects performed 5 cardio training sessions per week to reduce their body weight. The effect was a weight reduction of 3.9 ± 4.9 kg–5.2 ± 5.6 kg on average, depending on the size of the energy deficit. Gender differences were not observed [23]. Our study also confirms a similar weight loss in both the female and male groups in absolute terms, but it is worth noting that the men had a higher body weight at the beginning of the study than the women, and therefore the relative (percentage) weight reduction for the women was greater (17.4%) than for the men (14.7%). Expressing weight reduction in relative terms (percentages), in our opinion, is a much better way to determine the effectiveness and rate of excess weight reduction. There are studies showing a similar trend, i.e., better effects of weight reduction targeted programs in women. In a study by Sanal et. al., women reduced 4.3% of their body weight compared to men who reduced 3% of their body weight [24]. In the study by Gabriele et. al., women reduced 5.3% of body weight vs. 3.4% in men [25]. In opposition to these studies, there are publications showing that it is men who lose a greater percentage of body weight in studies targeting weight loss using changes in diet and physical activity levels [26,27]. Therefore, it is still unclear whether gender determines the magnitude and effectiveness of weight loss. Although men often show greater weight loss in absolute terms this is usually related to their greater baseline body weight. Additionally, it is worth noting that women in each study also significantly reduced their body weight. Currently, there is limited evidence saying that women and men should use different weight loss strategies. However, it is worth noting that strategies that combine dietary interventions with interventions that increase physical activity levels appear to be the most effective. The effectiveness of the dietary intervention combined with physical activity in the present study may have been due to the tailoring of the recommendations to the individual needs of the subjects and the continued support throughout the program. It seems that it is the combination of collaboration between the dietitian and trainer and active cooperation on the part of the participants that most contributes to the effectiveness of weight loss programs [28].

An important question addressed in this study was whether the age of the observed individuals affected the rate and effectiveness of weight reduction. It would seem that the older people get the more difficult it is to reduce body weight due to greater difficulty in acquiring new healthier eating habits or lower propensity to engage in increased physical activity such as trainings. Unfortunately, there are currently no studies that examine how age affects weight loss during weight loss therapy. Our results show that weight reduction occurs to a similar extent regardless of age. It is worth noting that this observation is unique in the literature. This may be related to the similar motivation to change one’s lifestyle among the participants regardless of age. The dietitian, in the course of cooperation with the participant, supported them appropriately in keeping to the diet and training goals, depending on the situation. Another hypothesis explaining the results is that dietary changes and physical activity were appropriately adjusted to each participant. Both the diet and the training were not imposed but adapted to the abilities of each participant. The proposed method of weight loss, associated with online communication, seems to be accepted by both genders and allows for similar results in different age groups. 

### Limitations of the Study

The study had some limitations. Due to the nature of the online intervention, we cannot confirm with certainty that the subjects adhered to the protocol 100 percent. Of course, in the course of the intervention, we checked the degree of program implementation. The study lasted 12 months, but it did not assess the degree of weight maintenance after the intervention. Due to the technological limitation, we also did not measure energy expenditure during exercise and throughout the day, which could have influenced the observed results. Furthermore, the study was unable to compare the mobile-based lifestyle intervention group with the offline intervention group to tell whether the online intervention was more effective than the offline one.

## 5. Conclusions

An important element of our study comparing gender differences in the perceptions of lifestyle change interventions among the subjects is the fact that men were comparable to women. We noted no differences in weight loss between men and women expressed in kilograms. However, women reduced weight more than men when the amount of reduction was assessed in relative terms (percentages). In other words, gender may be a differentiating factor in the magnitude of weight reduction in a long-term weight loss program. Our results also show that regardless of age, weight reduction occurs at a similar rate. This may be related to similar motivation to change one’s lifestyle among participants in the follow-up regardless of age. 

## Figures and Tables

**Figure 1 ijerph-19-12009-f001:**
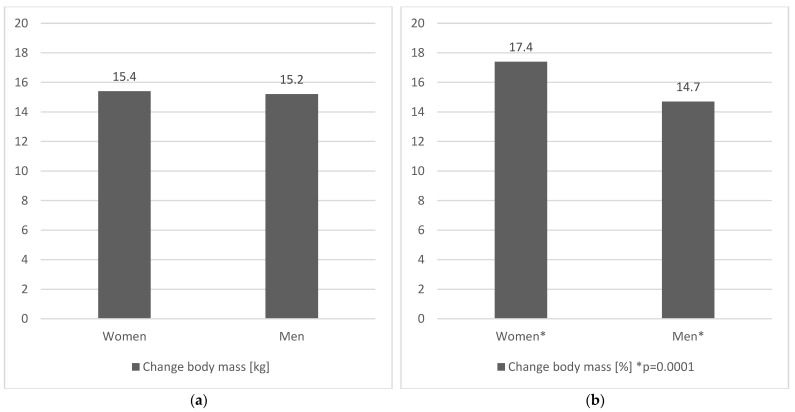
(**a**). Change in body weight (average) in men and women after 12 months expressed in absolute value [kg]. (**b**) Change in body weight (average) in men and women after 12 months expressed in relative value [%]. * Mann–Whitney U Test.

**Figure 2 ijerph-19-12009-f002:**
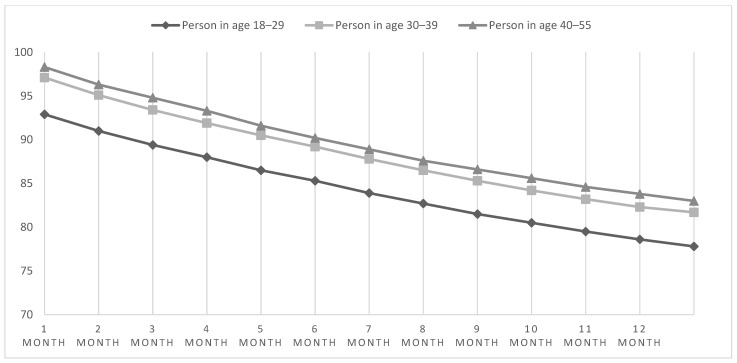
Mean rate of decline in body mass in the 3 age groups over 12 months [kg].

**Table 1 ijerph-19-12009-t001:** General characteristic of interventional diet.

Variable	Value
Caloric value [%]	85% TDEE
Proteins [g/kg body mass]	1.6 g
Fats in total [%]	25–35% of energy
Carbohydrates [%]	50–55% of energy
Fiber [g]	30–40
Saturated FA [%]	<5% of energy
Monosaturated FA [%]	14–26% of energy
Polysaturated FA [%]	4–6% of energy

TDEE—total daily energy expenditure; FA—fatty acids.

**Table 2 ijerph-19-12009-t002:** General characteristics of participants at the beginning of the intervention.

Variable	The Whole Group n = 400 Women = 210 Men = 190	
	Mean ± SD	Median (Min-Max)
Age [years]	33.42 ± 7.2	32 (19–55)
Height [m]	1.73 ± 0.09	1.73 (1.50–2.02)
Body mass [kg]	95.99 ± 17.01	95 (63–156.8)
BMI [kg/m^2^]	31.83 ± 4.77	30.9 (25.1–51.7)
BMR [kcal]	1950.1 ± 342	1911 (1323–3108)
PAL	1.49 ± 0.15	1.5 (1.2–2.0)
TDEE [kcal]	2883.9 ± 454.1	2837 (1984–4603)

BMI: body mass index; BMR: basal metabolic rate; PAL: physical activity level; TDEE: total daily energy expenditure.

**Table 3 ijerph-19-12009-t003:** Characteristics of women and men at the beginning of the intervention.

Variable	Men (n = 190)		Women (n = 210)		*p* *
	Mean ± SD	Median (Min-Max)	Mean ± SD	Median (Min-Max)	
Age [years]	33.43 ± 6.9	32 (19–55)	34.41 ± 7.45	31 (20–55)	0.48
Height [m]	1.80 ± 0.07	1.80 (1.57–2.02)	1.67 ± 0.06	1.68 (1.5–1.87)	0.00001
Body mass [kg]	103.6 ± 16	103 (64.8–156.8)	89.1 ± 14.8	87 (63–142)	0.00001
BMI [kg/m^2^]	31.9 ± 4.54	30.86 (25.1–48.9)	31.8 ± 4.98	31 (25.2–51.76)	0.54
BMR [kcal]	2156 ± 310	2147 (1360–3108)	1763 ± 251	1695 (1323–2898)	0.00001
PAL	1.48 ± 0.15	1.42 (1.2–2.0)	1.49 ± 0.14	1.5 (1.2–2.0)	0.07
TDEE [kcal]	3173 ± 410	3140 (2230–4603)	2621 ± 310	2592 (1984–3921)	0.00001
Trainings per week	3.13 ± 0.82	3 (1–7)	3.31 ± 0.84	3 (1–6)	0.01
Training time [min]	58.7 ± 17.96	60 (30–120)	55 ± 17.99	60 (30–120)	0.02
Training time per week [min]	185 ± 81.9	180 (90–330)	181.7 ± 76.7	180 (45–430)	0.59

* Mann–Whitney U Test.

**Table 4 ijerph-19-12009-t004:** Characteristics of participants at the beginning of the intervention by 3 age groups.

Variable	Person in Age 18–29 (n = 136) Women = 77 * Men = 59 *		Person in Age 30–39 (n = 156) Women = 75 * Men = 81 *		Person in Age 40–55 (n = 108) Women = 58 * Men = 50 *		*p* * 0.33
	Mean ± SD	Median (Min-Max)	Mean ± SD	Median (Min-Max)	Mean ± SD	Median (Min-Max)	*p* **
Height [m]	1.71 ± 0.09	1.72 (1.50–1.97)	1.74 ± 0.09	1.74 (1.5–2.02)	1.74 ± 0.08	1.74 (1.52–1.96)	0.21
Body mass [kg]	92.9 ± 15.6 ^a^	90.6 (65–133)	97.09 ± 17.9 ^b^	95 (63–156.8)	98.34 ± 16.9 ^b^	97.9 (67–142.6)	0.03
BMI [kg/m^2^]	31.17 ± 4.62	30.1 (25.1–51.76)	31.8 ± 4.84	30.86 (25.1–49.1)	32.7 ± 4.77	32.1 (25.2–51.4)	0.06
BMR [kcal]	1896 ± 313	1835 (1365–2793)	1979 ± 358	1953 (1323–3108)	1975 ± 351	1928 (1407–2994)	0.12
PAL	1.51 ± 0.14 ^a^	1.5 (1.3–2.0)	1.47 ± 0.13 ^b^	1.4 (1.2–1.9)	1.47 ± 0.14 ^b^	1.4 (1.2–2.0)	0.02
TDEE [kcal]	2859 ± 437	2793 (2029–4199)	2906 ± 481	2891 (1984–4603)	2882 ± 435	2832 (2133–4040)	0.64
Trainings per week	3.29 ± 0.86 ^a^	3 (1–6)	3.29 ± 0.86 ^a^	3 (1–7)	3.03 ± 0.8 ^b^	3 (1–4)	0.01
Training time [min]	56.9 ± 17.7	60 (30–120)	57.1 ± 20.5	60 (45–120)	56.1 ± 14.4	60 (30–90)	0.97
Training time per week [min]	189.4 ± 85	180 (45–400)	187.3 ± 83.4	180 (90–480)	169.9 ± 62.4	180 (60–360)	0.21

* Pearson’s Chi-square Test—Gender between group; ** Kruskal–Wallis one-way analysis of variance by ranks; ^a,b^ Kruskal–Wallis Test—Diference beetwen group.

## Data Availability

The data that support the findings of this study are available from the first author (J.W.) upon reasonable request.

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
