# Peer review of "Effect of Age and Gender on the Efficacy of a 12-Month Body Weight Reduction Program Conducted Online—A Prospective Cohort Study"

_ijerph, 2022, doi:10.3390/ijerph191912009_

Round 1
Reviewer 1 Report
The manuscript details a secondary analysis from a weight loss intervention study. In the first analysis, the effect of categories of BMI on weight loss was determined, with the finding that overweight, compared to obese subjects say greater weight loss (percentage). The study was a dietician -directed study where diet and exercise were tailored to the individual. The main finding of the study is that age had little effect, but there was an effect of gender, with women losing more weight (by percentage) than men.
Questions and Comments:
1. The study is based on a previous report and readers are asked to refer to the previous report for details of the intervention. The August 2022 study does describe the procedures of the intervention to some degree, but unless there are severe space limitations, the “respo” method should be described in this paper so that it stands alone.
2. Furthermore, even in the August 2022 paper, the methodology for individual participants is a bit unclear in that the diet and exercise are tailored to individuals. My biggest concern is that there may be differences in the tailored diets in men compared to women. The authors lay out the PAL, TDEE, and trainings for the participants at the beginning of the study, but then do not give us these data at the end or during the study. Is there any way to analyze if there were differences in trainings (exercise) or kcals between the participants during the intervention or at the end, which when stratified by gender may have contributed to the findings?
3. Authors refer to participants as “patients” which is misleading. They should be referred to as participants in the study.
4. Tables 5 and 6 may be better expressed as a bar graph. The tables could be supplementary data if a reader wants more detail.
5. The conclusion states that “We noted no difference in weight loss between men and women expressed in kilograms”…but this important information was not included in the abstract. Only the difference by percent body weight was noted in the abstract.
Author Response
|
Review Report Form |
Authors’ improvements |
|
The manuscript details a secondary analysis from a weight loss intervention study. In the first analysis, the effect of categories of BMI on weight loss was determined, with the finding that overweight, compared to obese subjects say greater weight loss (percentage). The study was a dietician -directed study where diet and exercise were tailored to the individual. The main finding of the study is that age had little effect, but there was an effect of gender, with women losing more weight (by percentage) than men. |
This is correct. In addition to the main conclusion, we also described other observations in our work. However, we agree with the maxim. |
|
The study is based on a previous report and readers are asked to refer to the previous report for details of the intervention. The August 2022 study does describe the procedures of the intervention to some degree, but unless there are severe space limitations, the “respo” method should be described in this paper so that it stands alone.
|
We add this paragraph in methodology:
The online intervention consisted of a 15% energy deficit diet and training (RESPO method). Respo method (from the word "responsiveness") is based on individual adjustment of the whole plan of losing weight. The Respo method is based on 4 fundamentals. The first is an individually tailored diet in terms of, among other things, taste preferences, number of meals or meal times. The second aspect of the method is a training plan adapted to the participants. The third foundation is constant contact with an online dietician, so that the participant could contact him at any time. The final aspect of this method is the emphasis on acquiring healthy eating habits. In the process of dietitian-participant collaboration, efforts were made to educate the patient as often as possible.
|
|
Furthermore, even in the August 2022 paper, the methodology for individual participants is a bit unclear in that the diet and exercise are tailored to individuals. My biggest concern is that there may be differences in the tailored diets in men compared to women. The authors lay out the PAL, TDEE, and trainings for the participants at the beginning of the study, but then do not give us these data at the end or during the study. Is there any way to analyze if there were differences in trainings (exercise) or kcals between the participants during the intervention or at the end, which when stratified by gender may have contributed to the findings? |
It is true that the diet as well as the training was tailored individually to each participant. However, both diet and training were based on the same foundations. The diet in each participant was universal in terms of: energy deficit, nutritional values, carbohydrate, fat and protein content. The training, on the other hand, in each case was a training adapted to beginners. Each training unit was composed so that the participant consumed a similar amount of energy on it.
In addition, all patients were monitored to perform spontaneous physical activity (NEAT) so that their PAL was at a similar level. |
|
Authors refer to participants as “patients” which is misleading. They should be referred to as participants in the study. |
Thank you for this comment, we have changed this in our work |
|
Tables 5 and 6 may be better expressed as a bar graph. The tables could be supplementary data if a reader wants more detail. |
Thank you for this comment, we add Figure 1a and Figure 1b in our paper. Also, we remove Table 5 and 6 from paper. We add this tables to supplementary material.
|
|
The conclusion states that “We noted no difference in weight loss between men and women expressed in kilograms” …but this important information was not included in the abstract. Only the difference by percent body weight was noted in the abstract. |
Thank you for this comment, we have changed this in abstract |

Reviewer 2 Report
The authors propose a study that analyzes the influence of gender and age on a 12 month body weight loss reduction program.
The study has several limitations, as stated by the authors themselves, first of all the difficulty in verifying the perfect adherence to the weight loss protocol of the sampled subjects.
However, the results are interesting, both that related to gender and that related to age.
I suggest some little clarification:
1) To make the methodological description of the study more complete, the authors should specify how the degree of program implementation was verified.
2) Since the study is based on a personalized program (diet and exercise) for each participant, I ask the authors if there is the possibility of verifying and analyzing any differences in training program or diet between the various experimental groups that may have contributed to the final results.
3) It would be interesting to analyze the weight loss of women within each age group to assess whether the pre or post menopausal period has an influence on the weight loss process.
4) In line 52-53 I think the sentence “In the EU, the percentage of overweight individuals is increased with weight” contains an error and that “weight” should be replaced with “age”.
5) I couldn't find the references 1-3 and 12-14, so I recommend authors to check and/or replace them.
Author Response
|
Review Report Form |
Authors’ improvements |
|
The authors propose a study that analyzes the influence of gender and age on a 12 month body weight loss reduction program. The study has several limitations, as stated by the authors themselves, first of all the difficulty in verifying the perfect adherence to the weight loss protocol of the sampled subjects. However, the results are interesting, both that related to gender and that related to age. |
Thank you for this comment. |
|
To make the methodological description of the study more complete, the authors should specify how the degree of program implementation was verified. |
We add this paragraph in methodology: |
|
Since the study is based on a personalized program (diet and exercise) for each participant, I ask the authors if there is the possibility of verifying and analyzing any differences in training program or diet between the various experimental groups that may have contributed to the final results. |
It is true that the diet as well as the training was tailored individually to each participant. However, both diet and training were based on the same foundations. The diet in each participant was universal in terms of: energy deficit, nutritional values, carbohydrate, fat and protein content. The training, on the other hand, in each case was a training adapted to beginners. Each training unit was composed so that the participant consumed a similar amount of energy on it.
In addition, all patients were monitored to perform spontaneous physical activity (NEAT) so that their PAL was at a similar level. |
|
It would be interesting to analyze the weight loss of women within each age group to assess whether the pre or post menopausal period has an influence on the weight loss process. |
In fact, it would be very interesting. This kind of observation would be an interesting extension of our research. However, a similar number of older women would need to be recruited to see this difference. In our study, there were only 16 women over the age of 50 |
|
In line 52-53 I think the sentence “In the EU, the percentage of overweight individuals is increased with weight” contains an error and that “weight” should be replaced with “age”. |
That’s right! We add changes. |
|
I couldn't find the references 1-3 and 12-14, so I recommend authors to check and/or replace them. |
We add new link/or replace these references. Thank you! |
